# Reconstitution of early paclitaxel biosynthetic network

Jack Chun-Ting Liu [1], Ricardo De La Peña [2], Christian Tocol[2] & Elizabeth S. Sattely [2,3] ✉

Paclitaxel is an anticancer therapeutic produced by the yew tree. Over the last two decades, a significant bottleneck in the reconstitution of early paclitaxel biosynthesis has been the propensity of heterologously expressed pathway cytochromes P450, including taxadiene 5α-hydroxylase (T5αH), to form multiple products. Here, we structurally characterize four new products of T5αH, many of which appear to be over-oxidation of the primary mono-oxidized products. By tuning the promoter strength for *T5αH* expression in *Nicotiana* plants, we observe decreased levels of these proposed byproducts with a concomitant increase in the accumulation of taxadien-5α-ol, the paclitaxel precursor, by three-fold. This enables the reconstitution of a six step biosynthetic pathway, which we further show may function as a metabolic network. Our result demonstrates that six previously characterized *Taxus* genes can coordinatively produce key paclitaxel intermediates and serves as a crucial platform for the discovery of the remaining biosynthetic genes.

Paclitaxel is an FDA-approved anti-cancer drug (Taxol®) used to treat breast, ovarian, and lung cancer. The drug stabilizes microtubules and in turn blocks cell cycle progression 1[1,2]. It was first discovered from the Pacific yew tree (*Taxus brevifolia*) through an anti-cancer screening program conducted by National Cancer Institute (NCI) in 1964[3]. Paclitaxel belongs to a class of diterpenoids from *Taxus* called taxanes, of which more than 500 structures have been reported[4]. Despite the success of paclitaxel in clinical applications, its limited availability due to low natural abundance (-0.004% in *Taxus* bark[2]) has prompted extensive research into alternative and sustainable production methods, including semi-synthesis from late-stage intermediates and cultivation of plant-cell cultures[5]. Refactoring plant biosynthetic pathways into microorganisms is a promising way to produce medically valued plant natural products like paclitaxel, and successful examples include antimalarial drug artemisinin[6] and anticholinergic drug scopolamine[7]. Nevertheless, the prerequisite for plant natural product production in microbial hosts is a complete understanding of the biosynthetic pathway. To date, despite extensive investigation over the past two decades, the full biosynthetic route to paclitaxel has not been elucidated.

Taxol biosynthesis is hypothesized to involve 19 *Taxus* enzymes, of which five are yet to be discovered (Fig. 1a, Supplementary Fig. 1)[8]. Elucidating the missing enzymes is hindered by the lack of access to intermediate substrates for functional characterization. The unknown biosynthetic enzymes are thought to act in the middle part of the pathway, as their substrates are notoriously difficult to obtain through synthesis or heterologous reconstitution (Supplementary Fig. 1). In the first committed step of paclitaxel biosynthesis, a diterpene cyclase taxadiene synthase (TS) converts geranyl geranyl pyrophosphate (GGPP) to taxadiene (**1**), which serves as the backbone of paclitaxel and related taxanes (Fig. 1a). Taxadiene 5α-hydroxylase (T5αH), a cytochrome P450 enzyme, is proposed to carry out the second step in the pathway, hydroxylating taxadiene (**1**) to yield taxadien-5α-ol (**2**) (Fig. 1a). However, the observed catalytic promiscuity of T5αH on heterologous expression has been a bottleneck in paclitaxel biosynthetic pathway reconstitution for nearly two decades. Expression of *T5αH* in different heterologous systems, such as *Escherichia coli*,[9–11] *Saccharomyces cerevisiae*[12,13] and *Nicotiana benthamiana*[14] (Supplementary Table 1), results in multiple oxidized taxadiene products. While up to 12 products have been reported[11], only three have been

[1]Department of Chemistry, Stanford University, Stanford, CA, USA. [2]Department of Chemical Engineering, Stanford University, Stanford, CA, USA. [3]Howard Hughes Medical Institute, Stanford University, Stanford, CA, USA. ✉e-mail: sattely@stanford.edu

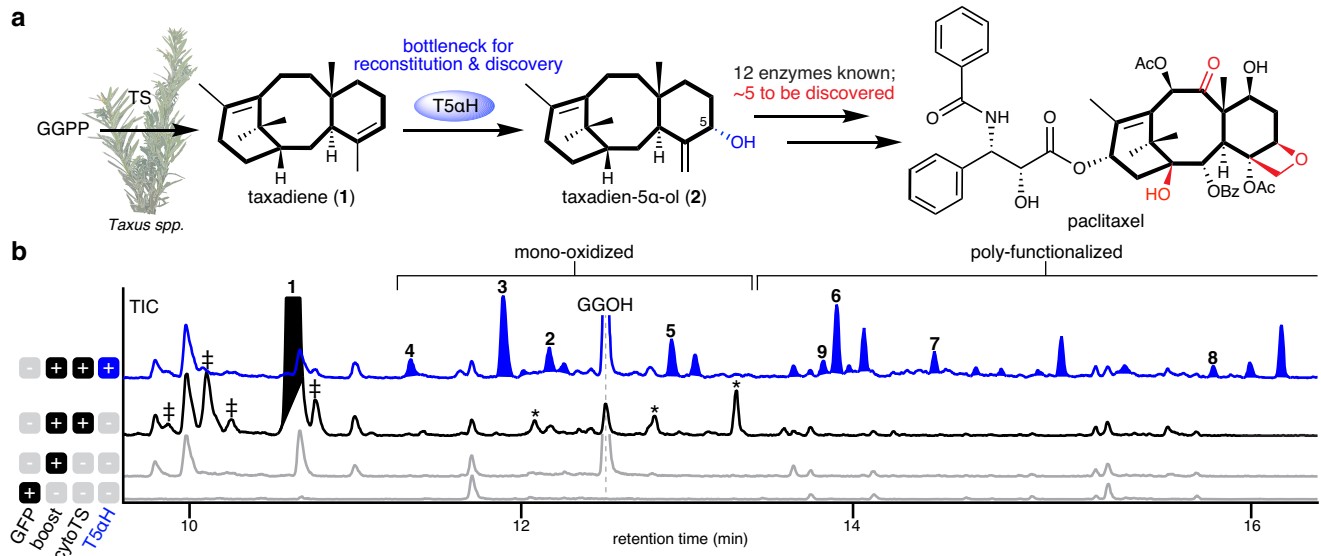

**Fig. 1 | T5αH catalytic promiscuity is a bottleneck for pathway reconstitution and discovery. a** In the proposed biosynthetic pathway for paclitaxel, taxadiene synthase (TS) converts geranyl geranyl pyrophosphate (GGPP) to taxadiene (**1**), which can be oxidized to taxadien-5α-ol (**2**) by taxadiene 5α-hydroxylase (T5αH, in blue circle). Texts relevant to T5αH and the functional group installed by T5αH are highlighted in blue. Functional groups in the paclitaxel structures where biosynthetic enzymes are yet to be discovered are highlighted in red. The core 6/8/6-membered ring scaffold is highlighted with bold bonds. Picture of *Taxus* plants is taken by Jack C-T. Liu and used with permission. **b** GCMS total ion chromatograms (TICs) of *Nicotiana benthamiana* leaves transiently expressing cytosolic *TS* (cyto*TS*, black trace) or cyto*TS* and *T5αH* (blue trace) under 35S promoter in the pEAQ-HT vector[27]. "Boost" indicates co-expression of truncated 3-hydroxy-3-methylglutaryl coenzyme-A reductase (t*HMGR*) and geranyl geranyl diphosphate synthase (*GGPPS*)

to increase cytosolic diterpenoid yield as previously reported[28] and are always used unless otherwise noted. Compound **3** and **4** represent OCT and iso-OCT, respectively. TICs of *N. benthamiana* expressing GFP or boost alone (gray traces) are shown as control backgrounds. Compound **5**–**8** are structurally characterized in this study. Taxadiene (**1**) peak is filled black (baseline tilted to indicate the co-eluting peak) and oxidized taxadiene peaks are filled blue. "‡" indicates minor products generated by TS, including verticillene, taxa-4(20),11(12)-dien (iso-taxadiene), and taxa-3(4),11(12)-diene[54], whose identities are not assigned here. "*" indicates mono-oxidized taxadiene generated in *N. benthamiana* presumably by endogenous enzymes. The peak for geranylgeraniol (GGOH), an endogenously hydrolyzed side-product of GGPP, is also indicated. Representative traces of three biological replicates are shown.

structurally characterized, namely, taxadien-5α-ol (**2**), rearranged products 5(12)-oxa-3(11)-cyclotaxane (OCT, **3**)[15] and 5(11)-oxa-3(11)-cyclotaxane (iso-OCT, **4**)[16]. In these heterologous systems, taxadien-5α-ol (**2**) is often a minor product among all oxidized taxadienes[2,10–15,17–19]. Taxadien-5α-ol (**2**) is generally regarded as the precursor to paclitaxel as **2** and its acetate ester can be transformed by *Taxus* microsomes to yield poly-oxidized products[20]. Conversely, there is still uncertainty regarding the involvement of other oxidized taxadienes products in paclitaxel biosynthesis in *Taxus*. While it has been proposed that oxidized taxadienes are merely the breakdown products of a reactive epoxide intermediate generated by T5αH[16,18,19], it has also been shown that oxidized taxadienes such as OCT (**3**) can be produced in *Taxus* plants when supplemented with a high level of taxadiene (**1**) and cannot be ruled out as paclitaxel precursors[19].

The observed catalytic promiscuity of T5αH hinders the reconstitution of paclitaxel biosynthesis as it reduces the metabolic flux toward taxadien-5α-ol (**2**) and multiple oxidized products potentially complicate product deconvolution. Attempts to minimize undesired oxidized taxadiene formation in microbial hosts have included engineering of T5αH and its cytochrome P450 reductase (CPR) partner[2,21] or through optimizing microbial growth conditions[2,21]. These approaches have failed to significantly increase the percentage of taxadien-5α-ol (**2**) among oxidized products, nor has permitted reconstitution of the biosynthesis by co-expressing the proposed downstream biosynthetic genes. Proposed downstream enzymes like taxadien-5α-ol *O*-acetyltransferase (TAT)[22], taxane 10β-hydroxylase (T10βH)[23], taxane 13α-hydroxylase (T13αH)[24] and 10-deacetylbaccatin III:10-*O*-acetyltransferase (DBAT)[25] have been individually characterized with purified substrates, but no reconstitution attempt has showed them all working in synchrony with TS and T5αH. To the best of our knowledge, the longest reconstitution efforts towards paclitaxel biosynthesis have

been the co-expression of *TS*, *T5αH*, *TAT* and *T10βH* through yeast-*E. Coli* co-cultures[17,26] and in *N. benthamiana*[14], respectively. However, in these studies, the proposed final product was not observed or produced in minor quantities insufficient for structural characterization by nuclear magnetic resonance (NMR). In this work, we determine the structures of four previously uncharacterized oxidized taxadiene products from T5αH, whose structures suggested that T5αH further oxidize primary products like taxadien-5α-ol (**2**) and OCT (**3**). We also demonstrate that over-expressed *T5αH* can directly oxidize **2**. Therefore, we propose overexpression of *T5αH* as one of the reasons for its observed catalytic promiscuity. By tuning the expression level of *T5αH* in *N. benthamiana*, we are able to increase the proportion of taxadien-5α-ol (**2**) among oxidized taxadienes and reduce the formation of confounding side-products. Using this optimized system, we reconstitute an early paclitaxel biosynthetic network with six characterized *Taxus* enzymes, where the final products are purified and structurally characterized by 2D-NMR. Accessing these early paclitaxel biosynthetic intermediates will enable future discovery of missing biosynthetic enzymes.

## Results

### Heterologous expression of *T5αH* with upstream pathway

To establish a baseline for taxadien-5α-ol (**2**) production prior work, we first transiently expressed *T5αH* in *N. benthamiana* via *Agrobacterium*-mediated transformation using the pEAQ-HT vector[27], where genes are expressed under the strong, constitutive 35S promoter to maximize protein production. *T5αH* was co-expressed simultaneously with a previously established taxadiene-enhancing system through re-localizing TS to cytosol (cyto*TS*) and overexpressing cytosolic mevalonate pathway genes (t*HMGR* and *GGPPS*)[28]. At least 36 oxidized products were observed by gas chromatography mass spectrometry

(GCMS) analysis of leaf extracts (Fig. 1b, Supplementary Fig. 2, Supplementary Table 2). All products are divided into two categories: "mono-oxidized taxadiene" and "poly-functionalized taxadiene" (Fig. 1b). Mono-oxidized taxadiene consists of products with a single oxidation identified based on their GCMS mass spectra. Poly-functionalized taxadiene includes mostly di- and tri-oxidized taxadienes indicated by their GCMS mass spectra, but also products with other modifications like ketone formation and hydrolysis (proposed based on GCMS mass spectra). All of these products are not unique to a plant host system: similarly, we observed 21 oxidized taxadienes when *TS* and *T5αH* were chromosomally integrated and co-expressed in *S. cerevisiae* (Supplementary Fig. 2)[29]. Among the oxidized products, only three have been thoroughly characterized – taxadien-5α-ol (**2**), OCT (**3**)[15] and iso-OCT (**4**)[16]. We synthesized taxadien-5α-ol (**2**) standard via an allylic oxidation on taxadiene (**1**) (Fig. 2a)[30], which can be easily purified from a *TS*-expressing yeast strain. With the standard, we confirmed taxadien-5α-ol (**2**) was heterologously produced in our *N. benthamiana* system as a minor product (Figs. 1b and 2b). The identity of OCT (**3**) was confirmed by NMR and mass spectrum (Supplementary Fig. 3) while the structural assignment of iso-OCT (**4**) was supported by its mass spectrum (Supplementary Fig. 3) and retention time earlier than **2** on GCMS (Fig. 1b) as previously reported[11,16]. Given the sheer number of uncharacterized products generated by T5αH in heterologous systems, it was unclear which compounds are true paclitaxel biosynthetic precursors, which are potentially involved in other non-paclitaxel biosynthetic pathways, and which are artifacts resulting from heterologous expression. Therefore, we set out to purify and structurally characterize oxidized taxadienes via large-scale infiltration of *N. benthamiana* plants and large-scale culturing of *S. cerevisiae* (Supplementary Table 3).

## Characterization of T5αH product profile

Mono-oxidized **5**, di-oxidized **6–7** (in acetylated form; result of co-expressing a TAT homolog, *TAX19*)[31] and di-oxidized taxadiene **8**, (Fig. 2c) were isolated and their structures were resolved through extensive 1D- and 2D-NMR analysis (Supplementary Tables 3–7). Structural assignment suggests that compound **5** contains a unique 6/7/6-membered ring scaffold and a 5,11-ether bond similar to the

structure of iso-OCT. We proposed that the formation of this scaffold is the result of two Wagner-Meerwein rearrangements that form C-C bonds between C-11/15 and C-12/16 (Supplementary Fig. 4). Compound **6** has a 4,19-ether bond and a 5-hydroxyl group similar to **2**. We found compound **7** to share the OCT (**3**) scaffold, but with an additional 9α-hydroxyl group. Our proposed structure of **8** has a 3,12-C-C linkage and 5,11-hydroxyl groups. We reasoned that **8** is a breakdown product of the 5,11-ether precursor [5(11)-oxa-3(12)-cyclotaxane], a hypothetical rearranged product produced alongside the formation of **3** and **4** (Supplementary Fig. 4). Notably, compounds **3–5** and **7–8** have distinct ether and/or C-C bonds in their structures that replace the usual C-11/12 double bond. Although we cannot rule out the possibility that these molecules represent true paclitaxel pathway intermediates, it seems unlikely given that the C-11/12 double bound is maintained in most taxanes isolated from *Taxus* plants[4], and no enzymes have been identified to have the ability to re-install this desaturation. We therefore assumed that taxadien-5α-ol (**2**) is the main precursor for paclitaxel biosynthesis among all the oxidized taxadienes generated by T5αH in heterologous systems.

Poly-functionalized products, especially di- and tri-oxidized taxadienes, may arise from over-oxidation of mono-oxidized taxadienes by T5αH. This is supported by previous characterization of 10-hydroxy-OCT as one of T5αH's minor products[32] and our structural characterization of **6** and **7** in this study. To test this hypothesis, we fed synthetic **2** to *N. benthamiana* leaves expressing *T5αH* and observed complete turnover of **2** to **9** (Fig. 2d). Compound **9** is proposed to be a di-oxidized product based on its mass spectra (Supplementary Table 2) and it accumulates as a minor product in our *T5αH*-expressing *N. benthamiana* system (Fig. 1b). The formation of over-oxidized products like **9** not only diverts the flux away from **2** but also complicates pathway reconstitution: when the next biosynthetic gene *TAT* is co-expressed with *TS* and *T5αH* in *N. benthamiana*, we observed two new peaks **10** and **11** with simultaneous disappearance of **2** and **9** (Fig. 2e), suggesting that TAT is able to accept both **2** and **9** as substrates. Additional introduction of downstream biosynthetic enzymes would likely result in a complex product profile where on-pathway paclitaxel intermediates are only present as minor products, as observed in previous reconstitution attempts[14,17,26]. We speculated that

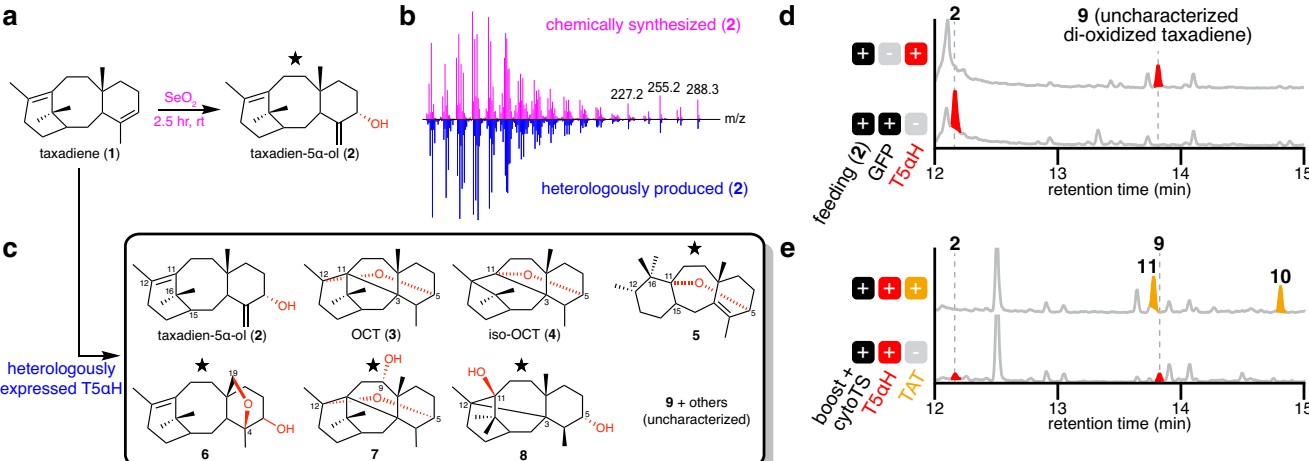

**Fig. 2 | Characterization of T5αH product profile. a** Synthesis of taxadien-5α-ol (**2**). Star indicates the structure is confirmed by NMR in this study. **b** GCMS mass spectra of **2** originated from chemical synthesis (magenta) and heterologous production in *N. benthamiana* (blue). **c** Structures of T5αH products when T5αH is heterologously expressed in *N. benthamiana*. Star indicates the structure is confirmed by NMR in this study. Oxidative modifications are shown in red. **d** GCMS total ion chromatograms (TICs) of *Nicotiana benthamiana* leaves transiently expressing *GFP* or *T5αH* with synthetic **2** fed into leaves three days post infiltration. Compound **9** is a di-oxidized

taxadiene whose structure is not characterized. **e** GCMS TICs of *Nicotiana benthamiana* leaves transiently expressing cytosolic diterpenoid boosting enzymes (boost) previously reported[28], cytosolic *TS* (cyto*TS*), *T5αH* and taxadien-5α-ol *O*-acetyltransferase (*TAT*). Two acetylated product **10** and **11** were observed with concurrent decrease of compound **2** and **9**, illustrating how the complexity of the product profile increases with additional downstream enzymes when T5αH side-products are present in the reconstitution system. Products of T5αH are colored red and those of TAT are colored orange.

over-oxidation is caused by the high levels of T5αH in the *N. ben-thamiana* over-expression system resulting in decoupled oxidation cycles and/or misfolded protein that retains undesirable activity. Given these results, we considered that lowering the production level of T5αH might mitigate over-oxidation and allow successful pathway reconstitution.

## Tuning T5αH transcription levels alleviates over-oxidation

The combination of *Agrobacterium* and pEAQ vector has been a robust system to transiently express non-native biosynthetic genes in *N. benthamiana* for plant biosynthetic pathway discovery over the last decade[27,33]. In the pEAQ vector, genes are expressed under a strong constitutive 35S promoter to maximize protein production[27,33]. To lower the T5αH production level, we identified two weaker constitutive promoters – ubiquitin-10 (UBQ10) and nopaline synthase (NOS) promoter – that resulted in 75% and 63% of the expression level observed with the 35S promoter, respectively (as estimated by GFP fluorescence

in *N. benthamiana*, Supplementary Fig. 5)[34]. We replaced the 35S promoter in pEAQ vector with either the UBQ10 or NOS promoter and tested T5αH activity in the new vectors when co-expressed with *TS*. Expression of *T5αH* under UBQ10 or NOS promoter greatly suppressed the formation of poly-functionalized taxadienes, and the highest taxadien-5α-ol (**2**) level was observed in the NOS system (Fig. 3a). Notably, taxadiene (**1**) was depleted in all three expression systems (Supplementary Fig. 6), suggesting that lower expression of *T5αH* does not comprise its utilization of **1**. By integrating the peak area of each oxidized product on the total ion chromatogram (TIC) and dividing by total integration area, we calculated the percentage for each product (Fig. 3b, Supplementary Fig. 6, Source Data) as an approximation of the product distribution (assuming all products have similar ionization efficiencies). The percentage of poly-functionalized products decreased significantly from 68% to 37% and 25% with UBQ10 or NOS promoters, respectively. Furthermore, synthetic **2** was no longer completely oxidized to **9** when fed into *N. benthamiana* leaves

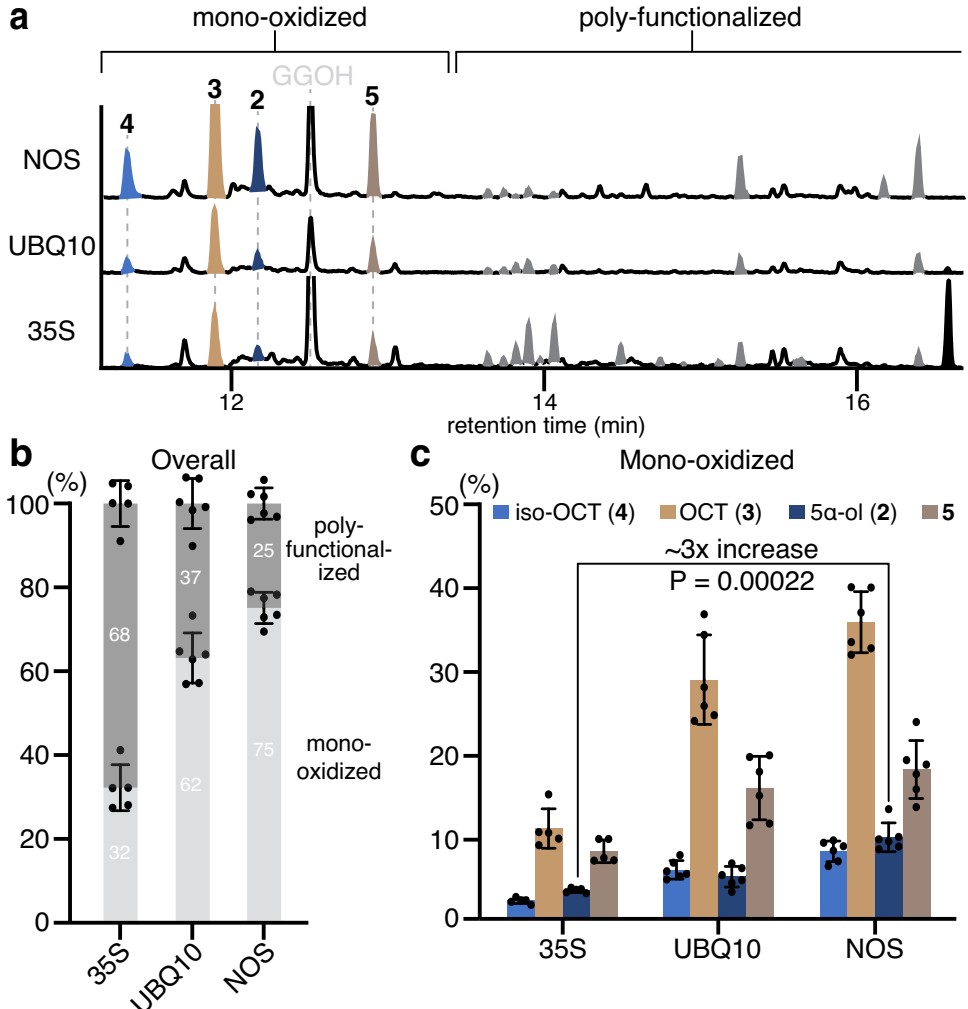

**Fig. 3 | Lowering *T5αH* expression level alleviates the formation of poly-functionalized taxadienes. a** GCMS total ion chromatograms (TICs) of *N. ben-thamiana* leaves transiently expressing cytosolic diterpenoid boosting genes previously reported[28], cytosolic taxadiene synthase, and *T5αH* where *T5αH* are expressed under either 35S, UBQ10, or NOS promoter. The scale for the y-axis is consistent across all traces, representing the absolute quantity of individual metabolites per unit dry weight of plant material. The peak for geranylgeraniol (GGOH), an endogenously hydrolyzed side-product of GGPP, is also indicated. Representative traces of six biological replicates are shown (Supplementary Fig. 6). **b** Bar graph showing the relative integrated peak area in percentage of all products,

categorized into mono-oxidized and poly-functionalized products. **c** Bar graph showing the relative integrated peak area in percentage for mono-oxidized tax-adienes **2**, **3**, **4**, and **5**. For **b** and **c**, total integrated area is normalized to 100% for each promoter based on the assumption that taxadiene production remains the same in each case. Data are shown as the mean ± standard deviation. *n* = 5 (NOS) or 6 (UBQ10 and NOS) biological, independent leaf samples. Statistical analyses were performed using a two-sided, unpaired Welch's *t*-test. Bar graphs were plotted using GraphPad Prism 9. Full table containing integrated peak area and percentage of each compound is provided as Source Data.

expressing (NOS)*T5αH* in contrast to (35S)*T5αH* (Supplementary Fig. 7), supporting our hypothesis that T5αH over-oxidation is due to its high expression level.

While the weaker promoters decreased the abundance of poly-functionalized products, the ratio between the four major mono-oxidized taxadienes (**2/3/4/5**) remained largely the same (Fig. 3c). Thus, mono-oxidized taxadienes like **3**–**5** may be formed due to incompatibility of T5αH with its microenvironment in heterologous systems compared to native plants. This phenomenon has been shown with other P450s where their activities are highly dependent on the membrane phospholipid composition[35] and the availability of native partner proteins (e.g., metabolon formation)[36]. With a more favorable microenvironment that mimics the native condition, it is possible that T5αH would generate **2** as the major mono-oxidized product. Future work to resolve the heterologous expression incompatibility is needed to increase the overall percentage of taxadien-5α-ol (**2**).

By tuning the expression level of *T5αH*, we achieved: (1) three-fold increase in the absolute level of taxadien-5α-ol (**2**) per leaf dried weight using the NOS promoter (Fig. 3a, c and Source Data), and (2) reduced formation of side-products, including poly-functionalized products and mono-oxidized products other than **2**–**5** (Fig. 3a, b), which minimizes potential confounding interactions with downstream biosynthetic enzymes. We next proceeded to reconstitute the early paclitaxel biosynthetic pathway using the engineered NOS promoter system.

## Reconstitution of early six-step paclitaxel biosynthetic pathway

Several *Taxus* enzymes proposed to be involved in early paclitaxel biosynthesis have been characterized, including cytochrome P450s T10βH[23] and T13αH[24] as well as acyltransferases TAT[22] and DBAT[25]. We attempted to reconstitute the early paclitaxel biosynthetic pathway by transiently co-expressing these genes in *N. benthamiana*. To thwart the potential over-oxidation by *Taxus* cytochrome P450s under 35S promoter, all *Taxus* P450s were expressed under the NOS promoter. When TAT was introduced to the engineered TS + (NOS)T5αH system, we observed the formation of 5α-acetoxytaxadiene (**11**) (Fig. 4) as the sole dominant product. In contrast to when TAT was introduced to the previous strongly expressed TS + (35S)T5αH system, compound **10**, the proposed acetylated product of **9**, was no longer formed as a side-product (Fig. 2e, Supplementary Fig. 8), demonstrating the success of the lower-expression system (Fig. 4). T10βH has been shown to efficiently oxidize 5α-acetoxytaxadiene (**11**) to 5α-acetoxytaxadien-10β-ol (**12**) when expressed in yeast[23], so we chose to introduce T10βH as the next enzyme for pathway reconstitution. When T10βH was introduced, a single new peak was observed coinciding with the disappearance of **11** (Fig. 4); we proposed this new product to be 5α-acetoxytaxadien-10β-ol (**12**) based on fragments of its mass spectrum (Supplementary Table 2). DBAT is an acyltransferase that acetylates 10-deacetylbaccatin III to baccatin III[25], and it has also been shown that purified *T. chinensis* DBAT is highly regio selective for C-10β *O*-acetylation on various partially deacetylated taxuyuannanine C (taxadien-2α,5α,10β-triacetoxy-14β-ol) substrates[37]. Given the high regio-selectivity of DBAT on less oxidized taxanes, we hypothesized that **12** can be transformed by DBAT if **12** indeed contains the 10β-hydroxyl group. The introduction of DBAT yielded a new peak with simultaneous disappearance of **12** (Fig. 4). Based on the mass spectrum of the new peak (Supplementary Table 2), we proposed this compound to be 5α,10β-diacetoxytaxadiene (**13**).

The early paclitaxel biosynthetic pathway is proposed to bifurcate from taxadien-5α-ol (**2**) by the action of TAT and T13αH (Supplementary Fig. 1)[24]. We decided to test T13αH together with all the enzymes (TS, T5αH, TAT, T10βH, and DBAT) to see if new products can be formed. We again saw the formation of a new product

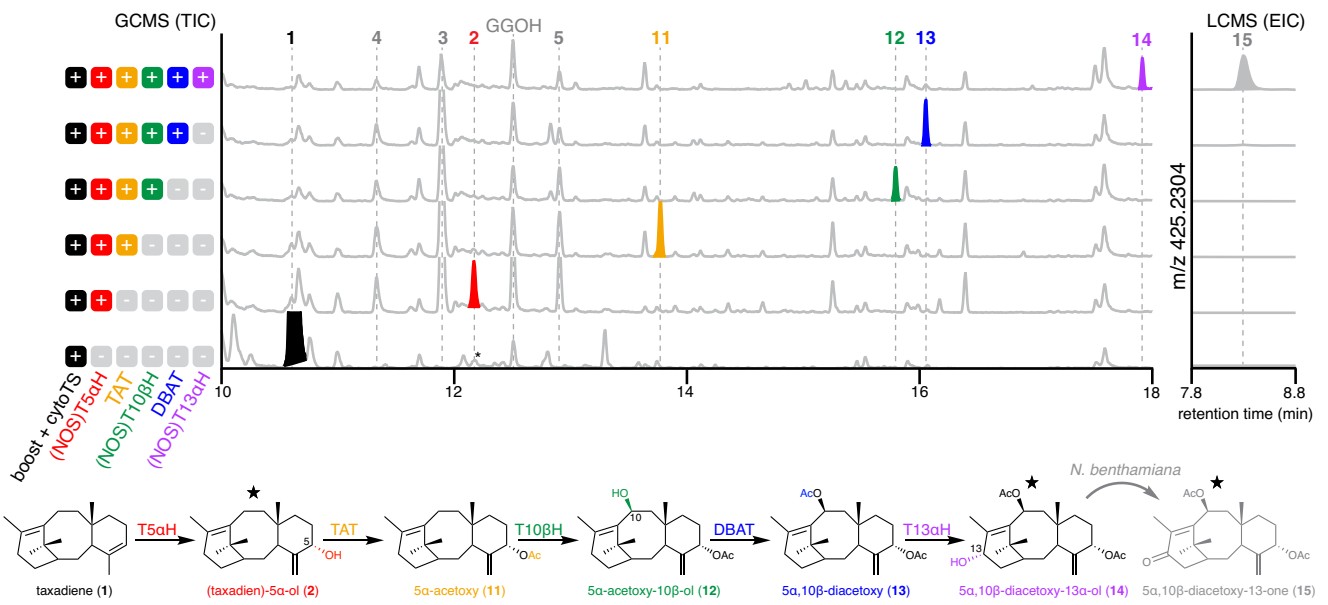

**Fig. 4 | Reconstitution of early paclitaxel biosynthetic pathway in *N. benthamiana*.** GCMS total ion chromatograms (TICs) and LCMS extracted ion chromatograms (EICs) of *N. benthamiana* leaf extracts after transiently expressing cytosolic diterpenoid boosting genes previously reported[28], cytosolic taxadiene synthase (cyto*TS*) and sequential addition of individual paclitaxel biosynthetic genes. The proposed pathway and intermediate structures are shown. Taxadiene 5-hydroxylase (T5αH) and its product are colored red; taxadien-5α-ol *O*-acetyl-transferase (TAT) and its product are colored orange; taxane 10β-hydroxylase (T10βH) and its product are colored green; 10-deacetylbaccatin III:10-*O*-acetyl-transferase (DBAT) and its product are colored blue; taxane 13α-hydroxylase (T13αH) and its product are colored purple. Cytochrome P450 *T5αH*, *T10βH* and *T13αH* are expressed under the NOS promoter, and all other genes, when not specified, are expressed under the 35S promoter. Structures of **11**–**13** are proposed based on the characterized functions of TAT, T10βH and DBAT in the literature[22,23,25]. "Taxadien" is omitted in the names of most compounds for simplicity. Star indicates that the structure is confirmed by NMR analysis. Asterisk indicates co-eluting compounds not related to taxanes. The peak for geranylger-aniol (GGOH), an endogenously hydrolyzed side-product of GGPP, is also indicated. Representative trace of three biological replicates is shown.

**14** and concurrent reduction in the amount of **13** (Fig. 4). Based on its MS spectrum, **14** was tentatively assigned as 5α,10β-diacetoxytaxadien-13α-ol (**14**) (Supplementary Table 2). While we initially used GCMS to detect hydrophobic molecules including **2**–**13**, as the taxadiene scaffold is decorated with more oxidations/acetylations and becomes more polar, we considered that untargeted liquid chromatography mass spectrometry (LCMS) may be more suitable for analysis. LCMS analysis of *N. benthamiana* extracts from leaves where all enzymes up to T13αH were expressed revealed 2 dominant products, one with the expected mass of **14** ([M+Na]$^+$ = 427.2460) while the other one (**15**) has an additional degree of unsaturation ([M+Na]$^+$ = 425.2304) (Fig. 4, Supplementary Fig. 9). To characterize the structures of **14** and **15**, large-scale infiltration and purification were carried out to yield **14** and **15** in 63 ug/g dry weight (DW) and 42 ug/g DW yield, respectively. Extensive 1D- and 2D-NMR elucidated their structures as 5α,10β-diacetoxytaxadien-13α-ol (**14**) and 5α,10β-diacetoxytaxadien-13α-one (**15**) (Supplementary Table 8). When **14** was fed into control *N. benthamiana* expressing *GFP*, it was partially oxidized to **15** (Supplementary Fig. 10). Therefore, we concluded that **15** results from oxidation by endogenous *N. benthamiana* enzymes rather than oxidation by T13αH (Fig. 4). The production and structural characterization of **14** in *N. benthamiana* provides the evidence for the first time that six *Taxus* biosynthetic enzymes (TS, T5αH, TAT, T10βH, DBAT, and T13αH) are working in synchrony to produce early paclitaxel intermediates.

## Metabolic network of early paclitaxel biosynthetic pathway

While we observed complete disappearance and appearance of what are proposed to be the substrate/product pairs with sequential introduction of biosynthetic enzymes from T5αH, TAT… to T13αH and illustrated the pathway as a linear series of steps (Fig. 4), it remains unclear in what order each enzyme acts. For example, it is possible that T13αH acts earlier in the pathway on **2** as previously reported[24] instead of directly oxidizing **13** to form **14**. To gain insight into the substrate preferences for these enzymes and further understand the reaction order of the pathway, we fed purified taxadien-5α-ol (**2**) into *N. benthamiana* leaves expressing different combinations of *TAT*, (NOS)*T10βH*, *DBAT*, and (NOS)*T13αH*. As expected, **2** can be efficiently transformed into **11**–**15** with sequential addition of TAT, T10βH, DBAT and T13αH (Fig. 5 exp.#1, 2, 6, 7 and 9), demonstrating that the new peaks accumulating in pathway reconstitution experiments (Fig. 4) indeed originate from **2**. Furthermore, we showed that **2** can be a direct substrate for either TAT or T13αH, but not T10βH (Fig. 5 exp.#1-4), which is consistent with previously proposed actions of TAT and T13αH that lead to early bifurcation in the pathway[20,24]. The oxidation of **2** by T13αH results in a product whose mass corresponds to taxadien-5α,13α-diol (**16**) (Fig. 5 exp.#4), and, similarly, masses for 5α-acetoxytaxadien-13α-ol (**18**) and 5α-acetoxytaxadien-10β,13α-diol (**20**) were observed in T13αH + TAT and T13αH + TAT + T10βH combination, respectively (Fig. 5 exp.#5 and 8). Products with masses corresponding to C-13 ketone

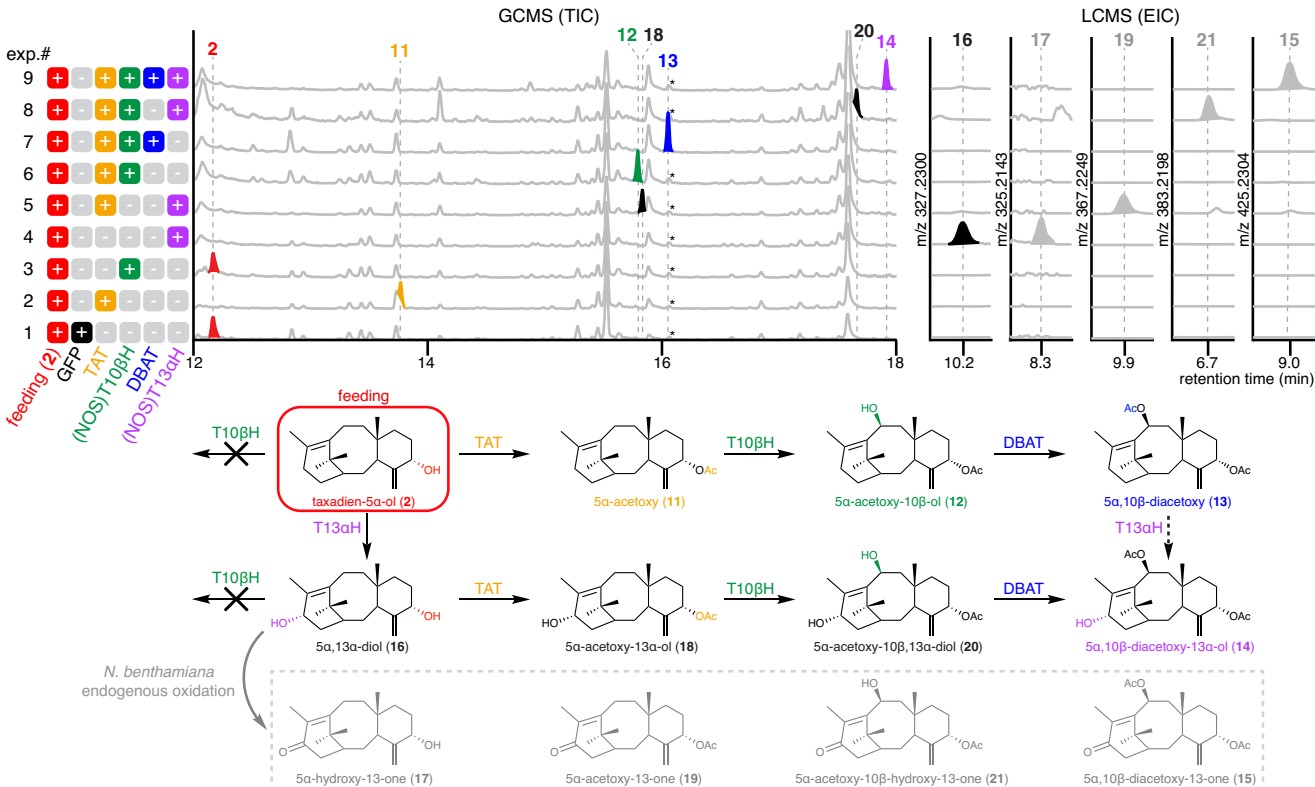

**Fig. 5 | Metabolic network of early paclitaxel biosynthesis.** GCMS total ion chromatograms (TICs) and LCMS extracted ion chromatograms (EICs) of plant extracts where taxadien-5α-ol (**2**) is fed into *N. benthamiana* leaves expressing different combinations of paclitaxel biosynthetic genes to examine all possible pathways. Taxadiene 5-hydroxylase (T5αH) and its product are colored red; taxadien-5α-ol *O*-acetyltransferase (TAT) and its product are colored orange; taxane 10β-hydroxylase (T10βH) and its product are colored green; 10-deacetylbaccatin III:10-*O*-acetyltransferase (DBAT) and its product are colored blue; taxane 13α-hydroxylase (T13αH) and its product are colored purple. Cytochrome P450 *T10βH* and *T13αH* are expressed under the NOS promoter, and all other genes are expressed under the 35S promoter. A retention time window of 0.5 min is shown for LCMS EIC. The proposed metabolic network and intermediate structures are shown. Structures **16**–**21** are proposed based on the characterized function of each enzyme in this work and the literature[22–25] as well as analysis of their mass spectra (Supplementary Table 2). "Taxadien" is omitted in the names of most compounds for simplicity. Dash arrow indicates reaction by T13αH that could potentially be involved in the metabolic network but for which we do not have direct evidence. Asterisk indicates co-eluting compounds not related to taxanes. The peak for geranylgeraniol (GGOH), an endogenously hydrolyzed side-product of GGPP, is also indicated.

5α-hydroxytaxadien-13-one (**17**), 5α-acetoxytaxadien-13-one (**19**), and 5α-acetoxy-10β-hydroxytaxadien-13-one (**21**) were also found in T13αH, T13αH + TAT and T13αH + TAT + T10βH combination, respectively (Fig. 5 exp.#4, 5 and 8). These C-13 ketone products likely result from *N. benthamiana* endogenous oxidation similar to the formation of **15** (Fig. 5, Supplementary Fig. 10).

When tested with purified substrates, it has been shown that recombinant T13αH oxidizes taxadien-5α-ol (**2**) efficiently, oxidizes 5α-acetoxytaxadiene (**11**) at 6% the conversion rate levels relative to **2**, and has no activity on 5α-acetoxytaxadien-10β-ol (**12**)[24]. Therefore, the formations of **18** and **20** likely go through **16** instead of direct oxidation by T13αH on **11** and **12**, respectively, as suggested in Fig. 5 scheme. Whether **13** is a direct substrate for T13αH to make **14** awaits further study. Compound 5α-acetoxytaxadien-10β-ol (**12**) and taxadien-5α,13α-diol (**16**) have been suggested as relevant biosynthetic intermediates in early paclitaxel biosynthesis as purified *Taxus* microsome produced **12** and **16** from exogenously supplied **11** and **2**, respectively[20]. Detection of intermediates downstream of **12** and **16** (**13**, **14**, **18** and **20**) supports a converging metabolic network with **14** as the final product, which unites the two pathway branches previously proposed at a common intermediate[20].

## Discussion

To increase production of the desired product in heterologous hosts, metabolic engineering often employs the strategy of overexpressing the biosynthetic genes or upstream precursor pathways[38,39]. However, over-expression of membrane-bound P450s in heterologous hosts often leads to problems such as decreased P450s solubility and imbalance in electron transfer[40–42]. Previous engineering efforts on T5αH has mostly focused on solving these issues by optimizing N-terminal sequence and CPR ratio[9,21]. Here, we show that overexpression of *T5αH* results in over-oxidation of the desired product (regardless of the heterologous host) while lowering the expression level of *T5αH* using NOS promoter proves to be beneficial for the reconstitution of the paclitaxel biosynthesis. By avoiding the compounding formation of multiple side-products, this strategy helps to streamline the biosynthesis of productive intermediates.

While we have successfully alleviated T5αH over-oxidation using engineered NOS promoter system, undesired mono-oxidized taxadienes like OCT (**3**), iso-OCT (**4**) and **5** still account for more than 65% of the accumulated taxanes (Source Data). Their relative ratios remain similar when switching promoters (Fig. 3c) and their accumulations persist with the introduction of downstream enzymes (Fig. 4). Notably, **3**–**5** or molecules with related structures have not been shown to accumulate in native *Taxus* species, suggesting **3**–**5** are artifacts of heterologous pathway reconstitution[4]. We proposed that these mono-oxidized taxadienes result from unfavorable conditions for T5αH in heterologous expression systems and/or missing *Taxus* partner proteins that would enable proper enzymatic function through metabolon formation[36]. Yet another possibility is that partial reconstitution of biosynthetic pathways results in accumulation of shunt products derived from unstable intermediates, which has been observed for the reconstitution of bacterial polyketide pathways[43]. In addition to the set of biosynthetic enzymes used for the reconstitution here, enzymes that directly reduce the production of mono-oxidized taxadienes might be required. Future efforts on the discovery of missing partner proteins and biosynthetic enzymes would help alleviate the observed catalytic promiscuity of T5αH. Furthermore, engineering of the heterologous expression system and T5αH may further enhance taxadien-5α-ol (**2**) production[42]. For example, addition of a cytochrome b₅ and lowering CPR expression ratio has shown to be critical for the optimal activity of CYP71AV1 in yeast for the production of artemisinin[44], and co-expression of heme biosynthetic genes can mitigate the heme-depletion stress posed by overexpression of P450s in yeast[45]. Our structural characterization of **5** and proposed mechanisms for the formation of mono-oxidized taxadienes **3**–**5** (Supplementary Fig. 4) could also serve as useful resources for rational protein engineering of T5αH to increase selectivity toward the production of **2**[10].

Distant phylogenetic relationships between native and heterologous hosts (in our case, *Taxus*, a gymnosperm, and *Nicotiana*, an angiosperm, plant) might cause issues in heterologous expression. Very few biosynthetic pathways from gymnosperm plants have been studied, and most involve many gymnosperm-specific P450s that are not well-characterized[46], for example, the biosynthesis of ginkgolide from *Ginkgo*[46,47]. Improper post-translational modifications could happen to heterologously expressed proteins in phylogenetically distant species, as demonstrated by the incorrect glycosylation of a plant P450 *SalSyn* when heterologously expressed in yeast[48]. In the future, an engineered heterologous system optimized for the expression of gymnosperm P450s will accelerate the study of biosynthesis in these ancient plants.

More than 500 structurally different taxanes have been isolated from *Taxus*, where combinations of different tailoring modifications (e.g., oxidation and acetylation) on the taxadiene scaffold have been reported[4]. The metabolic network we proposed here could potentially explain how such huge structural diversity arises from a handful of biosynthetic enzymes. Similarly, metabolic networks have been reported for the biosynthesis of many plant terpenoids, including *Arabidopsis* root triterpenoids[49], limonoids from *Citrus* and *Melia*[50], and cyclic AMP booster forskolin[51]. Together, these discoveries support the perspective that terpenoid specialized metabolism in plant is generally composed of inter-connected metabolic networks instead of linear pathways[52].

In summary, we have structurally characterized four oxidized taxadiene products from T5αH and confirmed overexpression as one of the mechanisms for undesired product formation. By using the lower-expression NOS promoter, we relieve the over-oxidation issue and increase the accumulation of taxadien-5α-ol (**2**) by three-fold. This allows successful reconstitution of an early paclitaxel biosynthetic network in a heterologous host involving TS, T5αH, TAT, T10βH, DBAT, and T13αH, concatenating the activities of these enzymes since their discoveries in the early 2000s. Our reconstitution effort of a relatively cleaner route to yield an isolatable level of 5α,10β-diacetoxytaxadien-13α-ol (**14**) will serve as a crucial platform for the discovery of downstream enzymes, including many of the P450s missing in the paclitaxel biosynthesis (Supplementary Fig. 1)[53].

## Methods

### Cloning of biosynthetic genes

The cloning of cytosolic diterpenoid boost, tHMGR and GGPPS, and cytosolic TS are described in previous study[28]. *Taxus* TS1, TS2, T5αH, TAT, T10βH, DBAT, T13αH, and TAX19 genes (Supplementary Table 9) were synthesized by Gen9 Bio (San Jose, CA) and amplified by PCR using Q5 High-Fidelity DNA Polymerase (New England BioLabs) and gene-specific primers from Integrated DNA Technologies (Supplementary Table 10). PCR amplicons were ligated with AgeI- and XhoI-(New England BioLabs) linearized pEAQ-HT vector[27] using HiFi DNA assembly mix (New England BioLabs). Constructs were transformed into 10-beta competent *E. coli* cells (New England BioLabs). Plasmid DNA was isolated using the QIAprep Spin Miniprep Kit (Qiagen) and sequence verified by Elim Biopharm.

### Transient expression of Taxus genes in N. benthamiana via Agrobacterium-mediated infiltration

pEAQ-HT plasmids containing the *Taxus* gene were transformed into *Agrobacterium tumefaciens* (strain GV3101) cells using the freeze-thaw method. Transformed cells were grown on LB-agar plates containing kanamycin and gentamicin (50 and 30 μg/mL, respectively; same for the LB media below), at 30 °C for 2 days. Single colonies were then picked and grew overnight at 30 °C in LB-Kan/Gen liquid media.

The overnight cultures were used to make DMSO stocks (7% DMSO) for long-term storage in the −80 °C fridge. For routine *N. benthamiana* infiltration experiments, individual *Agrobacterium* DMSO stocks were streaked out on LB-agar containing kanamycin and gentamicin and grew for 2 days at 30 °C. Patches of cells were scraped off from individual plates using 10 µL inoculation loops and resuspended in 1 mL of *Agrobacterium* induction buffer (10 mM MES pH 5.6, 10 mM MgCl$_2$ and 150 µm acetosyringone; Acros Organics) in individual 2 mL safe-lock tubes (Eppendorf). The suspensions were briefly vortexed to homogeneity and incubated at room temperature for 2 h. OD600 of the individual *Agrobacterium* suspensions were measured, and the final infiltration solution where OD600 = 0.2 for each strain was prepared by mixing individual strains and diluting with the induction buffer. Leaves of 4-weeks old *N. benthamiana* were infiltrated using needleless 1 mL syringes from the abaxial side. Leaf-6, 7, and 8 (numbered by counting from the bottom) of *N. benthamiana* were used, and each experiment was tested on different plants and leaves as three biological replicates (for example, experiment 1 is tested on plant-1 leaf-6, plant-2 leaf-7, and plant-3 leaf-8).

## Metabolite extraction of N. benthamiana leaves

*N. benthamiana* leaf tissue 5-days post *Agrobacterium* infiltration was collected using a leaf disc cutter 1 cm in diameter and placed inside a 2 mL safe-lock tube (Eppendorf). Each biological replicate consisted of 4 leaf discs from the same leaf (approximately 40 mg fresh weight). The leaf discs were flash-frozen and lyophilized overnight. To extract metabolites, ethyl acetate (ACS reagent grade; J.T. Baker) 500 µL was added to each sample along with one 5 mm stainless steel bead. The samples were homogenized in a ball mill (Retsch MM 400) at 25 Hz for 2 min. After homogenization, the samples were centrifuged at 18,200 × *g* for 10 min. For GCMS samples, the supernatants were filtered using hydrophobic PTFE filters with 0.45 µm pore size (Millipore) and transferred to 50 µL glass inserts placed in 2 mL vials. For LCMS samples, 100 µL of supernatants were transferred to a new microcentrifuge tube and evaporated to dryness under N$_2$. The samples were reconstituted in 100 µL of acetonitrile (HPLC grade; Fischer Chemical), filtered using hydrophilic PTFE filters with 0.45 µm pore size (Millipore) and transferred to LCMS vials.

## Large-scale fermentation of yeast culture

Starter yeast cultures (*TS*- or *TS + T5αH*-expressing; Supplementary Table 11) were grown in 3 mL of YPD media (10 g/L yeast extract, 20 g/L peptone, dextrose 40 g/L) in several 15 mL tubes overnight at 30 °C. The overnight cultures were used to inoculate (1:100 ratio) a total of 1–2 L of YPD media divided into multiple 500 mL or 1 L flasks where the liquid volume didn't exceed 20% of the flask volume. The inoculated cultures were grown for 2 days at 30 °C to saturation. The saturated cultures were centrifuged at 2700 × *g* for 10 min and the resulting pellet was resuspended in YPG media (10 g/L yeast extract, 20 g/L peptone, galactose 40 g/L) to induce the expression of *TS*. The induced cultures were grown at 20 °C for 72 h with additional 10% (V:V) galactose (40% in water) supplemented at 24 and 48 h. The induced cultures were combined for the purification of taxadiene (*TS*-expressing) or mono-oxidized taxadienes (*TS + T5αH*-expressing).

## Purification of taxadiene (1) from large-scale yeast culture

Equal volume of ethyl acetate (ACS reagent grade; J.T. Baker) and approximate 10 g of 0.5 mm glass beads (BioSpec) were added to 400 mL of induced *TS*-expressing yeast culture in a 2 L flask. The flask was shaken at 20 °C overnight. The culture was centrifuged at 2700 × *g* for 10 min. The pellet was discarded, and the supernatant (organic and aqueous layers) was extracted with 400 mL of ethyl acetate twice in a 2 L separatory funnel. Additional centrifugation of the supernatant might be helpful for the separation of the two phases. Organic phases

were combined and dried using rotary evaporation. Flash chromatography was carried out using a 7 cm diameter column loaded with 20 g of P60 silica gel (SiliCycle) with hexane (HPLC grade; VWR) as the mobile phase. Isocratic elution (100% hexane) was performed and eluent was collected in 5 mL fractions. Fractions were analyzed by GCMS and those containing taxadiene were combined and dried by rotary evaporation to yield 15.5 mg of taxadiene (**1**) as colorless oil (39 mg/L). Iso-taxadiene was co-purified with **1** in a 1:10 ratio as indicated by NMR peak integration. The $^1$H and $^{13}$C NMR data of taxadiene (**1**) are reported in Supplementary Figs. 11–13.

## Synthesis of taxadien-5α-ol (2)

The synthesis of taxadien-5α-ol (**2**) is adapted from a previously described method[30]. Tert-BuOOH (21.3 µL of 98% purity, 0.114 mmol; Sigma–Aldrich) was added to a solution of SeO$_2$ (3.3 mg, 0.030 mmol; Strem) in CH$_2$Cl$_2$ (50 µL; Sigma–Aldrich) and stirred for 0.5 h. Subsequently, purified taxadiene (**1**) (15.5 mg, 0.0570 mmol) in CH$_2$Cl$_2$ (200 µL) was added to the solution and allowed to stir for 2.5 h at room temperature. The resulting yellow solution was concentrated on a rotary evaporator under reduced pressure to yield yellow crude oil. Flash chromatography was carried out using a 2 mL glass Pasteur pipette loaded with 200 mg of P60 silica gel (SiliCycle) with hexane (HPLC grade; VWR) as the initial mobile phase. The column was first eluded with 100% hexane to yield unreacted **1** (21% recovery), then with hexane-diethyl ether (Fischer Chemical) in a ratio of 5:1 to yield 2.3 mg of an over-oxidized ketone product (TLC R$_f$ = 0.63 when eluded with hexane-Et$_2$O 5:1 and stained with KMnO$_4$; 14% isolated yield) and 1.8 mg of **2** (TLC R$_f$ = 0.36 when eluded with hexane-Et$_2$O 5:1 and stained with KMnO$_4$; 11% isolated yield). Fractions were collected in 200 µL volume. The $^1$H NMR data of synthetic taxadien-5α-ol (**2**) are reported and compared to **2** heterologously produced in *E. coli* to confirm its identity (Supplementary Figs. 14–15).

## Feeding taxadien-5α-ol (2) to N. benthamiana transiently expressing Taxus genes

*Taxus* genes were expressed using the *Agrobacterium*-mediated infiltration method described above. At 3-days post *Agrobacterium* infiltration, a 10 mM stock solution Taxadiene 5α-ol (**2**) in DMSO was diluted 100-fold with water to prepare 100 µm **2** solutions and infiltrated into *N. benthamiana* leaves expressing *Taxus* genes. Approximate 150 µL of the 100 µm **2** solutions was used per leaf to yield a circle with a diameter around 3 cm, which was marked for reference. After 18–24 h, the leaves were harvested by cutting along the marked area with a scissor and lyophilized overnight in pre-weighed 2 mL safe-lock tubes (Eppendorf). GCMS and LCMS samples were prepared following the aforementioned methods, with the exception of using ethyl acetate at a ratio of 20 µL per mg of dried-weight material.

## GCMS analysis

GCMS samples were analyzed using an Agilent 7820 A gas chromatography system coupled to an Agilent 5977B single quadrupole mass spectrometer. Data were collected with Agilent Enhanced MassHunter and analyzed by MassHunter Qualitative Analysis B.07.00. Separation was carried out using an Agilent VF-5HT column (30 m × 0.25 mm × 0.1 µm) with a constant flow rate of 1 ml/min of helium. The inlet was set at 280 °C in split mode with a 10:1 split ratio. The injection volume was 1 µl. Oven conditions were as follows: start and hold at 130 °C for 2 min, ramp to 250 °C at 8 °C/min, ramp to 310 °C at 10 °C/min and hold at 310 °C for 5 min. Post-run condition was set to 320 °C for 3 min. MS data were collected with a mass range 50–550 m/z and a scan speed of 1562 u/s after a 4-min solvent delay. The MSD transfer line was set to 250 °C, the MS source was set to 230 °C and the MS Quad was set to 150 °C. The mass spectra of all compounds are reported in Supplementary Table 2.

## LCMS analysis

LCMS samples were analyzed using an Agilent 1260 HPLC system coupled to an Agilent 6520 Q-TOF mass spectrometer. Data were collected with Agilent MassHunter Workstation Data Acquisition and analyzed by MassHunter Qualitative Analysis 10.0. Separation was carried out using a Gemini 5 μm NX-C18 110 Å column (2 × 100 mm; Phenomenex) with a mixture of 0.1% formic acid in water (A) and 0.1% formic acid in acetonitrile (B) at a constant flow rate of 400 μL/min at room temperature. The injection volume was 2 μl. The following gradient of solvent B was used: 3% 0–1 min, 3%–50% 1–2 min, 50%–97% 2–12 min, 97% 12–14 min, 97%–3% 14–14.5 min and 3% 14.5–21 min. MS data were collected using electrospray ionization (ESI) on positive mode with a mass range 50–1200 m/z and a rate of 1 spectrum/s. The ESI source was set as follows: 325 °C gas temperature, 10 L/min drying gas, 35 psi nebulizer, 3500 V VCap, 150 V fragmentor, 65 V skimmer, and 750 V octupole 1 RF Vpp. The mass spectra of all compounds are reported in Supplementary Table 2.

## Large-scale infiltration of N. benthamiana

For large-scale infiltration, infiltration solution containing *Agrobacterium* (OD600 = 0.6 per strain) was prepared as described above with the following modifications: each *Agrobacterium* strain was grown in 1 L of LB-Kan/Gen for 2 days to saturation and harvested by centrifugation at $2700 \times g$ for 10 min. Typically, 30–60 *N. benthamiana* (5–6 weeks old) were used in one experiment. Whole plants were submerged in 1.5 L infiltration solution and infiltrated under vacuum for at least 2 min. After 6–8 days post infiltration, leaves from infiltrated *N. benthamiana* were chopped into small pieces and lyophilized for 2–3 days.

## Extraction and purification of taxanes from N. benthamiana and yeast

Compounds **3, 5**, acetylated **6**, acetylated **7, 14,** and **15** were isolated from large-scale infiltration of *N. benthamiana*. Compound **1** and **8** were isolated from large-scale fermentation of yeast. Combinations of biosynthetic genes, scale (number of plants or volume of yeast culture) and yield are summarized in Supplementary Table 3. Dried *N. benthamiana* materials were extracted with 2 L ethyl acetate (ACS reagent grade; J.T. Baker) in a 4 L flask for 48 h at room temperature with constant stirring. Extracts were filtered using vacuum filtration and dried using rotary evaporation. *TS + T5αH*-expressing yeast culture (4*1 L) were directly extracted with 500 mL ethyl acetate (ACS reagent grade; J.T. Baker) in 2 L flasks with the addition of 0.5 mm glass beads (BioSpec) and shaken 20 °C overnight. The culture was centrifuged at $2700 \times g$ for 10 min. The pellet was discarded, and the supernatant (organic and aqueous layers) was extracted with an equal volume of ethyl acetate twice in a 2 L separatory funnel. Additional centrifugation of the supernatant might be helpful for the separation of the two phases. Organic phases were combined and dried using rotary evaporation.

Two rounds of chromatography were used to isolate compounds of interest from the extracts of *N. benthamiana* and yeast. Chromatography conditions for each compound are summarized in Supplementary Table 3. In short, the first chromatography was performed using a 7-cm-diameter column loaded with P60 silica gel (SiliCycle) and using hexane (HPLC grade; VWR) and ethyl acetate as the mobile phases. The second chromatography was carried out on an automated Biotage Selekt system with a Biotage Sfar C18 Duo 6 g column using Milli-Q water and acetonitrile as the mobile phases. Fractions were analyzed by GCMS or LCMS to identify those containing the compound of interest. Desired fractions were pooled and dried using rotary evaporation (first round) or lyophilization (second round). Purified products were analyzed by NMR.

## NMR analysis of purified compound

CDCl$_3$ (Acros Organics) was used as the solvent for all NMR samples unless otherwise specified. $^1$H, $^{13}$C, and 2D-NMR spectra were acquired using either a Varian Inova 500 MHz or 600 MHz spectrometer at room temperature using VNMRJ 4.2, and the data were processed and visualized on MestReNova v14.3.1. Chemical shifts were reported in ppm downfield from Me$_4$Si by using the residual solvent (CDCl$_3$) peak as an internal standard (7.26 ppm for $^1$H and 77.16 ppm for $^{13}$C chemical shift). Spectra were analyzed and processed using MestReNova version 14.3.1-31739. The NMR data for compounds **1, 2, 3, 5**, acetylated **6**, acetylated **7, 8, 14,** and **15** are provided in Supplementary Figs. 11–54 and the $^1$H/$^{13}$C assignments for **5**, acetylated **6**, acetylated **7, 8, 14,** and **15** are shown in Supplementary Tables 4–8.

## Reporting summary

Further information on research design is available in the Nature Portfolio Reporting Summary linked to this article.

## Data availability

The raw NMR free induction decay (FID) data of individual compounds have been deposited in the Natural Products Magnetic Resonance Database (np-mrd.org) with the following ID: **1** (NP0332339), **2** (NP0332341), 3 (NP0332444), **5** (NP0332420), acetylated **6** (NP0332441), acetylated **7** (NP0332421), **8** (NP0332442), **14** (NP0332342) and **15** (NP0332340); the processed NMR data are available in Supplementary Figs. 11–54 and Supplementary Tables 4–8. The integrated peak areas of individual compounds under the three promoters used for the plotting of Fig. 3 are provided in Source Data. Sequences of the biosynthetic enzymes used in this study are derived from GenBank with the following accession numbers: taxadiene synthase (U48796 and AY364470), T5αH (AY289209), TAT(AF190130), T10βH (AF318211), DBAT (AF456342), T13αH (AY056019), which are also listed in Supplementary Table 9. Source data are provided with this paper.

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

## Acknowledgements
We would like to thank the Keasling lab (University of California, Berkeley) for providing the engineered yeast strains and, specifically, Michael Belcher and Graham Hudson (University of California, Berkeley) for discussion on fermentation protocol of the yeast strains. We also want to thank Stephen Lynch (Stanford University) for his helpful discussion on NMR structural elucidation. The author is thankful to all Sattely lab members for constructive feedback during the writing of this manuscript. This work is supported by NIH R01 AT010593 (E.S.S).

## Author contributions
J.C-T.L. and R.D.L.P. cloned the biosynthetic genes. R.D.L.P. developed the GCMS condition for taxane detection. J.C-T.L. and C.T. performed large-scale purification and structural characterization. J.C-T.L. constructed the plasmids for tuning expression level. J.C-T.L. and E.S.S. analyzed the data and wrote the manuscript. E.S.S. supervised the work.

## Competing interests
The authors declare no competing interests.
