## [Peer Review File · Nature Communications]

REVIEWER COMMENTS

Reviewer #1 (Remarks to the Author):

The manuscript by Liu et al. describes the reconstitution of the early paclitaxel biosynthetic network. Paclitaxel is one of the most famous natural products with an outstanding significance in medicine and science history, yet there are still major gaps in our understanding of its biosynthetic pathway. A key challenge has been that one of the enzymes in the early pathway, T5aH, produces complex product mixtures with many undesired byproducts in addition to the required pathway intermediate taxadien-5a-ol. The current manuscript now addresses this problem in the common heterologous plant host *Nicotiana benthamiana*. The authors demonstrate that using the strong 35S promoter facilitates the formation of overoxidised products, whereas weaker promoters lead to less of the undesired overoxidation products. Four of the overoxidation products were also structurally characterised. This discovery was then employed to reconstitute the first six biosynthetic steps towards paclitaxel, allowing the authors to show that the biosynthesis represents a metabolic grid rather than a linear pathway and enabling isolation and NMR confirmation of the putative intermediate 5a,10b-diacetoxy-taxadiene-13a-ol (14) and the shunt product 15.

While the metabolic engineering aspect of this manuscript is relatively short and simple, being based only on the comparison of three different promoters, the resulting change in the metabolic profile is highly relevant, as it leads to 4x increase of taxadien-5a-ol levels and reduction of byproduct levels. Also, the authors do an excellent job combining natural product chemistry, synthetic biology and plant biochemistry. The structure elucidation of four new products of T5aH was carried out in a fully convincing way and provides important information for further enzyme engineering approaches. Also, the manuscript provides very useful information about the order of biosynthetic transformations and substrate specificities of pathway enzymes in paclitaxel biosynthesis. Overall, in my opinion this is an excellent multidisciplinary study that provides crucial insights into one of the most important biosynthetic pathways and will be significant to many researchers interested in this pathway, plant biochemistry and metabolic engineering. Before publication, several minor weaknesses should be addressed:

- Some of the compounds grouped under di- and tri-oxidised taxadienes don't seem to be true double or triple oxidation products. Most importantly, compound 8 is a hydrolysis product (as correctly shown by the authors in Extended Data Fig. 4) and therefore on the same oxidation level as OCT (3) and other mono-oxidised taxadienes. I would suggest that the authors change the terminus and refer to the number of oxygen atoms instead of the number of oxidation events to solve this problem, unless they have clear evidence (e.g. from MS) which compounds are truly di- and tri-oxidised. This aspect might also be discussed more in the manuscript.

- The authors seem to have mixed up esters and ethers. Throughout the manuscript, there are multiple mentions of esters, esterification etc. (e.g. Extended Data Fig. 4), when only an ether (R-O-R) is shown instead of an ester (R-CO-OR). Please check every occurrence of "ester" and correct if necessary.

- Compound nomenclature: some compound names used by the authors sound incorrect to me, for example taxadien-5a-acetoxy (11), taxadien-5a,13a-diacetoxy (13). I would argue that these names are

not IUPAC-conform, as acetoxy describes a substituent and is not a proper suffix. Instead, other names like 5 α -acetoxytaxadiene or taxadien-5 α -yl acetate should be preferred. Taxadien-5 α -ol-13-one (17) also sounds incorrect, a compound cannot have two suffixes. Please carefully check all compound names again.

- In the discussion you write: "Yet another possibility is that partial reconstitution of biosynthetic pathways result in accumulation of shunt products derived from unstable intermediates, which has been observe for the reconstitution of bacterial polyketides. 44" but is that argument not ruled out by your experimental work, where you reconstitute at least the first six steps of the pathway and still observe these shunt products? Please also check language ("result[s]", "observe[d]").

- p. 4: "were isolated and their structures were resolved" maybe you could provide the yields (e.g. ug/g dry weight) here to give the reader an idea of the product levels?

- Figure 1: please show full stereochemistry. The tertiary bridgehead carbons lack stereochemical information.

- p. 14: "Chemical shifts were reported in ppm downfield from Me 4 Si by using the residual solvent peak as an internal standard." please provide ¹H and ¹³C values that you used for referencing

- Supplementary Table 3: please check C18 column conditions and description of mobile phase. Are assignments A = acetonitrile and B = water really correct or swapped? Did you really increase the proportion of water in the gradients during C18 chromatography?

Language etc.:

- "protein expression" and similar phrases are used several times throughout the manuscript, but strictly speaking proteins are not expressed but rather produced. Only genes are expressed. Please check usage of expression.

- p. 4: "retention time earlier than 2 on GCMS (Figure 1B) as previously reported. 11 16" please check reference format, missing comma or dash

- Figure 2 legend: "was converted into to a "

- "5 α ,10 β -diacetoxy-taxadien[e]-13 α -ol (14)" no e after dien

- p. 8: "each additional enzyme added and illustrate the pathway as a linear series of steps (Figure 4)" this sentence reads confusing to me, please check language

- p. 11: "these discoveries supports the perspective that" support?

- p. 12: "freezed" frozen

- p. 13: "weighted" weighed

- p. 14: "palette" do you mean pellet?

- Extended Data Fig. 5: change "Fluorescent" to "Fluorescence" in title and legend

- Extended Data Fig. 9 and 10: compound names are incomplete, missing "taxadien"

- Supplementary Table 1: "fugiperd" should be "frugiperda"
- Supplementary Table 12: some parts of the table might be cut off - should this page be in landscape orientation?

Reviewer #2 (Remarks to the Author):

The manuscript of Liu and co-coworkers describes the optimization of the reconstitution of the early steps of paclitaxel biosynthesis in *Nicotiana benthamiana*. It is worth noting that the production of this highly valuable anticancer drug remained limited in heterologous organisms so far. This notably results from the hijacking of biosynthetic intermediates leading to the formation of overoxidized products that could not supply the biosynthetic pathway up to the ultimate and desired compound. In the present work, the authors addressed the formation of these undesired compounds by focusing on the taxadiene 5 α -hydroxylase (T5H), one of the key enzymes of the Paclitaxel biosynthetic route, known to synthesize multiple products. By reconstituting the early steps of the biosynthetic path of this drug and fine-tuning the specific expression of T5H by using promoters of distinct strengths, the authors solved, at least partially, the problem of the formation of the multi-oxidized T5H products they have also identified for several of them. This resulted in a four-fold increase of taxadien-5 α -ol. Overall, the experiments have been well conducted and the results are convincing. The manuscript is well written and elegantly highlights the main discovery of this work. Minor comments and questions are appended below:

- The use of the NOS promoter compared to 35S led to a (well-known) decrease of gene expressions and in T5H case, a reduction of the over-oxidized products. Can similar results be obtained by decreasing the OD of the agrobacterium strain solution expressing this gene compared to those expressing the remaining genes?
- Do all P450s from this pathway catalyze product over-oxidation? Why using the NOS promoter to express all P450s? Is there any undesired decrease of activity and product formation?
- Is this result obtained by fine-tuning T5H expression transposable to other heterologous models such as yeast?

To all reviewers:

We thank all reviewers for thoroughly reading the manuscript and providing constructive feedback. Below, we have addressed the raised questions and concerns, point-by-point. We have also colored major text changes in red in the revised manuscript and copied relevant texts here for ease of reading.

Reviewer #1 (Remarks to the Author):

The manuscript by Liu et al. describes the reconstitution of the early paclitaxel biosynthetic network. Paclitaxel is one of the most famous natural products with an outstanding significance in medicine and science history, yet there are still major gaps in our understanding of its biosynthetic pathway. A key challenge has been that one of the enzymes in the early pathway, T5aH, produces complex product mixtures with many undesired byproducts in addition to the required pathway intermediate taxadien-5a-ol. The current manuscript now addresses this problem in the common heterologous plant host *Nicotiana benthamiana*. The authors demonstrate that using the strong 35S promoter facilitates the formation of overoxidised products, whereas weaker promoters lead to less of the undesired overoxidation products. Four of the overoxidation products were also structurally characterised. This discovery was then employed to reconstitute the first six biosynthetic steps towards paclitaxel, allowing the authors to show that the biosynthesis represents a metabolic grid rather than a linear pathway and enabling isolation and NMR confirmation of the putative intermediate 5a,10b-diacetoxy-taxadiene-13a-ol (14) and the shunt product 15.

While the metabolic engineering aspect of this manuscript is relatively short and simple, being based only on the comparison of three different promoters, the resulting change in the metabolic profile is highly relevant, as it leads to 4x increase of taxadien-5a-ol levels and reduction of byproduct levels. Also, the authors do an excellent job combining natural product chemistry, synthetic biology and plant biochemistry. The structure elucidation of four new products of T5aH was carried out in a fully convincing way and provides important information for further enzyme engineering approaches. Also, the manuscript provides very useful information about the order of biosynthetic transformations and substrate specificities of pathway enzymes in paclitaxel biosynthesis. Overall, in my opinion this is an excellent multidisciplinary study that provides crucial insights into one of the most important biosynthetic pathways and will be significant to many researchers interested in this pathway, plant biochemistry and metabolic engineering. Before publication, several minor weaknesses should be addressed:

- Some of the compounds grouped under di- and tri-oxidised taxadienes don't seem to be true double or triple oxidation products. Most importantly, compound 8 is a hydrolysis product (as correctly shown by the authors in Extended Data Fig. 4) and therefore on the same oxidation level as OCT (3) and other mono-oxidised taxadienes. I would suggest that the authors change the terminus and refer to the number of oxygen atoms instead of the number of oxidation events to solve this problem, unless they have clear evidence (e.g. from MS) which compounds are truly di- and tri-oxidised. This aspect might also be discussed more in the manuscript.

We agreed with the reviewer's concern that the name "di- and tri-oxidized" might be misleading as they contain compounds outside these categories. The categorization for di- and tri-oxidized

taxadiene was initially designated based on the mass spectrum of major peaks. However, as the reviewer pointed out, the structures we have elucidated show that di- and tri-oxidized categories include hydrolysis products (e.g. compound **8**), and potentially compounds with other modifications beyond oxidations (e.g. ketone formation). Therefore, we decided to combine the two categories (di- and tri-oxidized) and called it “poly-functionalized product”. In this way we hope to capture all modifications beyond mono-oxidation, including hydrolysis, di-, tri-oxidation. The “mono-oxidized product” remains the same as we have either confirmed their structure (compound **2-5**) or observed parent mass corresponding to mono-oxidized taxadiene based on their GCMS mass spectra. We have revised the figures and text accordingly.

“All products are divided into two categories: “mono-oxidized taxadiene” and “poly-functionalized taxadiene” (**Figure 1b**). Mono-oxidized taxadiene consists of products with a single oxidation identified based on their GCMS mass spectra. Poly-functionalized taxadiene includes mostly di- and tri-oxidized taxadienes indicated by their GCMS mass spectra, but also products with other modifications like ketone formation and hydrolysis (proposed based on GCMS mass spectra).”

- The authors seem to have mixed up esters and ethers. Throughout the manuscript, there are multiple mentions of esters, esterification etc. (e.g. Extended Data Fig. 4), when only an ether (R-O-R) is shown instead of an ester (R-CO-OR). Please check every occurrence of "ester" and correct if necessary.

We have checked all occurrences of “ester” and “ether” in the manuscript and corrected the mix-up when needed.

For example:

“Structural assignment suggests that compound **5** contains a unique 6/7/6-membered ring scaffold and a 5,11-ether bond similar to the structure of iso-OCT.”

Extended Data Fig. 4 (now Supplementary Fig. 4) “ether formation”

- Compound nomenclature: some compound names used by the authors sound incorrect to me, for example taxadien-5 α -acetoxy (**11**), taxadien-5 α ,13 α -diacetoxy (**13**). I would argue that these names are not IUPAC-conform, as acetoxy describes a substituent and is not a proper suffix. Instead, other names like 5 α -acetoxytaxadiene or taxadien-5 α -yl acetate should be preferred. Taxadien-5 α -ol-13-one (**17**) also sounds incorrect, a compound cannot have two suffixes. Please carefully check all compound names again.

We agreed that some of the compounds were not correctly named following IUPAC conformation. We have revised the names of these compounds (short name shown in parentheses) as below:

5 α -acetoxytaxadiene (**11**) (5 α -acetoxy)

5 α -acetoxytaxadien-10 β -ol (**12**) (5 α -acetoxy-10 β -ol)

5 α ,10 β -diacetoxytaxadiene (**13**) (5 α ,10 β -diacetoxy)

5 α ,10 β -diacetoxytaxadien-13 α -ol (**14**) (5 α ,10 β -diacetoxy-13 α -ol)

5 α ,10 β -diacetoxytaxadien-13-one (**15**) (5 α ,10 β -diacetoxy-13-one)

taxadien-5 α ,13 α -diol (**16**) (5 α ,13 α -diol)

5 α -hydroxytaxadien-13-one (**17**) (5 α -hydroxyl-13-one)

5 α -acetoxytaxadien-13 α -ol (**18**) (5 α -acetoxy-13 α -ol)

5 α -acetoxytaxadien-13-one (**19**) (5 α -acetoxy-13-one)

5 α -acetoxytaxadien-10 β ,13 α -diol (**20**) (5 α -acetoxy-10 β ,13 α -diol)

5 α -acetoxy-10 β -hydroxytaxadien-13-one (**21**) (5 α -acetoxy-10 β -hydroxy-13-one)

The short names, where "taxadien" is omitted are used in most figures (e.g. Figure 4, 5, and Supplementary Fig. 9 & 10) for ease of reading, as all compounds share the same taxadiene scaffold and we only want to highlight the major enzymatic modifications on this scaffold.

- In the discussion you write: "Yet another possibility is that partial reconstitution of biosynthetic pathways result in accumulation of shunt products derived from unstable intermediates, which has been observe for the reconstitution of bacterial polyketides. 44" but is that argument not ruled out by your experimental work, where you reconstitute at least the first six steps of the pathway and still observe these shunt products? Please also check language ("result[s]", "observe[d]").

While we have reconstituted six steps of the pathway, this might still be an incomplete set of all the Taxol biosynthetic enzymes, and additional upstream enzymes (e.g. enzymes that directly help clean up T5aH shunt products) might still be required. We have added a sentence in the main text to clarify this point.

"In addition to the set of biosynthetic enzymes used for the reconstitution here, enzymes that directly reduce the production of mono-oxidized taxadienes might be required."

- p. 4: "were isolated and their structures were resolved" maybe you could provide the yields (e.g. ug/g dry weight) here to give the reader an idea of the product levels?

The yields are provided in Supplementary Table 3, which is referred to in the quoted sentence.

"were isolated and their structures were resolved through extensive 1D- and 2D-NMR analysis (Supplementary Table 3-7)"

- Figure 1: please show full stereochemistry. The tertiary bridgehead carbons lack stereochemical information.

The structures of taxadiene, taxadien-5 α -ol, and paclitaxel in **Figure 1** have been updated to show full stereochemistry on the two bridgehead carbons.

- p. 14: "Chemical shifts were reported in ppm downfield from Me 4 Si by using the residual solvent peak as an internal standard." please provide ¹H and ¹³C values that you used for referencing

We have provided the values to the sentence.

"7.26 ppm for ¹H and 77.16 ppm for ¹³C chemical shift"

- Supplementary Table 3: please check C18 column conditions and description of mobile phase. Are assignments A = acetonitrile and B = water really correct or swapped? Did you really increase the proportion of water in the gradients during C18 chromatography?

The solvent assignments for A and B were incorrectly placed. The correct assignment (A =water, B = acetonitrile) has been updated.

Language etc.:

- "protein expression" and similar phrases are used several times throughout the manuscript, but strictly speaking proteins are not expressed but rather produced. Only genes are expressed. Please check usage of expression.

We have checked all occurrences of "expression" (including co-expression, over-expression, expression level...etc) and corrected the subjects to *gene names* (represented in italic), and the usages of "protein expression" have been corrected to "protein production".

For example:

"*T5αH* was co-expressed simultaneously with a previously established taxadiene-enhancing system..."

"By tuning the expression level of *T5αH*..."

"In the pEAQ vector, genes are expressed under a strong constitutive 35S promoter to maximize protein production.^{27,34} To lower the *T5αH* production level..."

- p. 4: "retention time earlier than 2 on GCMS (Figure 1B) as previously reported. 11 16" please check reference format, missing comma or dash

This has been addressed.

- Figure 2 legend: "was converted into to a "5α,10β-diacetoxy-taxadien[e]-13α-ol (14)" no e after dien

This typo has been corrected.

- p. 8: "each additional enzyme added and illustrate the pathway as a linear series of steps (Figure 4)" this sentence reads confusing to me, please check language

We have revised this sentence to "with sequential introduction of biosynthetic enzymes from *T5αH*, *TAT*... to *T13αH* and illustrated the pathway as a linear series of steps (Figure 4)".

- p. 11: "these discoveries supports the perspective that" support?

This typo has been corrected.

- p. 12: "freezed" frozen

This typo has been corrected.

- p. 13: "weighted" weighed

This typo has been corrected.

- p. 14: "palette" do you mean pellet?

This typo has been corrected.

- Extended Data Fig. 5: change "Fluorescent" to "Fluorescence" in title and legend

These typos have been corrected.

- Extended Data Fig. 9 and 10: compound names are incomplete, missing "taxadien"

In the figures (Figure 4, 5, Extended Data Fig. 9 and 10), "taxadien" is omitted for simplicity and this is explained in the caption. Please also see above our response to question 3.

- Supplementary Table 1: "fugiperd" should be "frugiperda"

This typo has been corrected.

- Supplementary Table 12: some parts of the table might be cut off - should this page be in landscape orientation?

The table has been updated to avoid seemingly being cut off.

Reviewer #2 (Remarks to the Author):

The manuscript of Liu and co-covers describes the optimization of the reconstitution of the early steps of paclitaxel biosynthesis in *Nicotiana benthamiana*. It is worth noting that the production of this highly valuable anticancer drug remained limited in heterologous organisms so far. This notably results from the hijacking of biosynthetic intermediates leading to the formation of overoxidized products that could not supply the biosynthetic pathway up to the ultimate and desired compound. In the present work, the authors addressed the formation of this undesired compounds by focusing on the taxadiene 5 α -hydroxylase (T5H), of the key enzymes of the Paclitaxel biosynthetic route, known to synthesize multiple products. By reconstituting the early steps of the biosynthetic path of this drug and fine-tuning the specific expression of T5H by using promoters of distinct strengths, the authors solved, at least partially, the problem of the formation of the multi-oxidized T5H products they have also identified for several of them. This resulted in a four-fold increase of taxadien-5 α -ol. Overall, the experiments have been well conducted and the results are convincing. The manuscript is well written and elegantly highlights the main discovery of this work. Minor comments and question are appended below:

- The use of the NOS promoter compared to 35S led to a (well-known) decrease of gene expressions and in T5H case, a reduction of the over-oxidized products. Can similar result be obtained by decreasing the OD of the agrobacterium strain solution expressing this gene compare to those expressing the remaining genes?

We have indeed tried to reduce gene expression by using a lower OD of the T5aH agrobacterium strain. While we can observe a similar result (less over-oxidized products) by lowering OD, we noticed a lower overall product level, which is emphasized by the accumulation of taxadiene precursor, as compared to complete conversion of taxadiene in the case of 35S or NOS promoter (as shown in **Supplementary Fig. 6**). Furthermore, the levels of over-oxidized products, especially those that are relatively low to begin with, go down below the detection limits of the GCMS. As a result, it's difficult to determine the relative ratio of products and compare results directly to those from using agrobacterium at normal OD 0.2.

We speculate that, at lower T5aH agrobacterium OD (< 0.2), a significant percentage of the tobacco cells would not get infected nor express T5aH, thus those cells will only accumulate taxadiene. To maximize taxadiene turnover and reduce T5aH expression level at the same time, we chose to use NOS-controlled T5aH agrobacterium at normal OD to make sure all tobacco cells are infected while all infected cells express T5aH at lower levels.

- Do all P450s from this pathway catalyze product over oxidization? Why using of the NOS promoter to express all P450s? Is there any undesired decrease of activity and product formation?

For the other two P450s we used in the manuscript (T10 β H and T13 α H), they both produce several other minor products when expressed under the 35S promoter. Expression under the NOS promoter greatly mitigates the issue and results in the relatively cleaner transformation from compound **11** to **14**. These data (T10 β H and T13 α H with the NOS promoter) are shown in main text Figure 4. We are currently studying the underlying cause of this catalytic promiscuity

and trying to understand if all *Taxus* P450s show the same behavior or only those in the earlier pathway (T5 α H, T10 β H and T13 α H) do.

- Is this result obtained by fine-tuning T5H expression transposable to other heterologous model such as yeast?

Tuning expression of T5aH have been conducted in *E. coli* (citation 21: "Overcoming heterologous protein interdependency to optimize P450-mediated Taxol precursor synthesis in *Escherichia coli*" in the manuscript), where they found a 5-copy T5aH plasmid construct showed a higher oxygenated taxanes yield than a 10-copy T5aH plasmid construct. However, they did not report the specific increase in yield for taxadien-5 α -ol. In addition, fine-tuning expression level of plant P450s and their reductase partners has been shown to be critical for the production of artemisinin in yeast (citation 45: "High-level semi-synthetic production of the potent antimalarial artemisinin"). Both papers are cited and discussed in the main text. Given these examples, we believe it's fair to say that our result can be transportable to other heterologous systems.

Figure 3 update:

In compliance with the journal formatting guideline, we have repeated experiments in **Figure 3** with larger sample sizes ($n = 6$) to show individual data points and perform statistical analysis, and provided original results in **Source Data**. In doing so, we observed a three-fold increase ($P = 0.00022$) in taxadidn-5 α -ol production instead of the four-fold increase we previously reported with a smaller sample size. Thus, we have changed the claimed increase of taxadidn-5 α -ol production throughout the manuscript to three-fold.

REVIEWERS' COMMENTS

Reviewer #1 (Remarks to the Author):

I thank the authors for their corrections. All my points have been sufficiently addressed and I can now recommend the manuscript for publication.

Reviewer #2 (Remarks to the Author):

In the revised version of their manuscript, the authors have addressed all the concerns raised by my reviewing. I've appreciated reading this manuscript and want to congratulate the authors for their work.